# The Use of Cavitron Ultrasonic Surgical Aspirator for High-Risk Neuroblastoma with Image-Defined Risk Factors in Children

**DOI:** 10.3390/children10010089

**Published:** 2023-01-02

**Authors:** Luca Pio, Florent Guérin, Cristina Martucci, Helene Martelli, Frédéric Gauthier, Sophie Branchereau

**Affiliations:** 1Department of Surgery, St. Jude Children’s Research Hospital, Memphis, TN 38105, USA; 2Paediatric Surgery Department, Hôpital Bicêtre APHP, Paris Saclay University, 91190 Paris, France

**Keywords:** neuroblastoma, CUSA, cavitron ultrasonic surgical aspirator, high-risk neuroblastoma, image-defined risk factors

## Abstract

**Aim of the study**: The cavitron ultrasonic surgical aspirator (CUSA) has gained popularity in adult surgical oncology, but its application in children is limited to liver surgery and neurosurgical procedures. The complete resection of neuroblastoma with image-defined risk factors (IDRFs) is still considered one of the most difficult procedures to achieve in pediatric surgical oncology, with a high morbidity rate and potential risk of intraoperative mortality. The aim of our study is to describe the application of ultrasonic dissection in neuroblastoma with IDRFs. **Methods**: A retrospective study was performed, analyzing patients operated on from 2000 to 2018. Patient characteristics, resection completeness, and postoperative surgical and oncology outcomes were analyzed. **Main results:** Twenty-six patients with high-risk neuroblastoma and IDRFs were operated on in the study period with a CUSA. A complete macroscopic resection was performed in 50% of patients, while the other half was operated on with minimal residual (<5 mL). Six post-operative complications occurred without the need for surgery (Clavien–Dindo < 3). The overall survival was 50%, with a median follow-up of 69.6 months (5.6–140.4). **Conclusions:** The application of the CUSA in neuroblastoma with IDRFs can be considered an effective and safe alternative technique to achieve a radical resection.

## 1. Introduction

Neuroblastoma (NB) represents the most frequent extracranial solid tumor in children, with a heterogeneous presentation in different anatomic sites and a typical behavior towards vascular structures, rarely infiltrating the adventitious tunic, but displacing and surrounding it [1] (Figure 1).

Vascular involvement currently represents the most challenging scenario in children undergoing surgical resection for NB. In the last several years, the International Neuroblastoma Risk Group Staging System (INRGSS), based on NB vascular characteristics, was developed to guide the timing for surgery, identifying image-defined risk factors (IRDFs) [2,3].

In 2007, Kiely described the most frequently applied technique for the surgical resection of NB with vascular involvement, describing tumor incision with piecemeal removal and a vascular dissection to the tunica media with a cold scalpel [4].

Despite the introduction of several technological advances, including robotic surgery and other perioperative tools to guide surgical resection such as three-dimensional modeling, the vascular dissection of neuroblastic tumors with IDRFs remains a challenging procedure with potential life-threatening events and organ injury. A radical resection represents a controversial topic in those tumors with favorable biology, while complete resection has been shown to be related to better oncological outcomes [5,6].

The use of the Cavitron Ultrasonic Surgical Aspirator (CUSA) was first described by Hodgson et al. in 1979 [7] and extensively applied over the years in adult and pediatric neurosurgery due to its ability to respect vascular tunica during tumor dissection.

In pediatric literature, the application of the CUSA has only been reported by Loo in 1988 as a feasible technique for neuroblastoma resection in a case series of five patients; nevertheless, this approach was never further reported [8]. 

This article describes the technical detail of the CUSA for pediatric neuroblastoma resection with IDRFs to achieve a complete macroscopic resection.

## 2. Technical Procedure

Under general anesthesia, children were positioned in dorsal decubitus. A transverse supraumbilical incision was performed instead of a median laparotomy, due to the transversal abdominal development during childhood. In the case of preoperative IDRFs involving the porta hepatis and supra-mesenteric aorta, a reverse Mercedes incision was performed to obtain sufficient vascular control.

Retroperitoneal space access was obtained previa right/left colon mobilization, starting dissection from the avascular Toldt line with bowel peritoneal mobilization. The aorta and inferior cava veins were isolated in their cranial and caudal extra tumoral location with a colored silicone vascular loop in order to control any eventual intraoperative hemorrhage.

Tumor dissection was started by combining monopolar and bipolar electrosurgery until the identification of the vascular encasement locations. En-bloc resection was avoided to prevent any organ failure.

The neuroblastoma capsule was initially opened using monopolar electrosurgery, and then the CUSA^®^ Excel (Integra LifeSciences, Princeton, NJ, USA) was applied for the parenchyma dissection.

Tumor fragmentation is achieved by the vibrational energy of the surgical tip by an alternating voltage at 36 kHz; the suction system of the instrument, with a vacuum pump that provides a maximum pressure of 640 mmHg, allows for continuous tissue contact with the tip, enhancing tumor fragmentation. An irrigation system is supplied by an adjustable pump connected to the handpiece with an adjustable flush of saline or lactated ringer’s solution; the continuous flux from the irrigation hole of the tip provides the clearance of the adjacent suction hole and adequate aspiration. 

The energy created by the ultrasonic waves fragments parenchymal cells, owing to their high-water content, and selectively spares blood vessels and ducts (such as bile ducts), due to their poor water and high collagen content. 

Tissue selectivity is related to the quantity of collagen and/or elastin that increase tissue strength and subsequent fragmentation power. In addition, different tissue strengths (parenchymal/vascular structures) perceived by the CUSA tip provide tactile feedback to the surgeon, who can feel the difference in tissues encountered during dissection.

Different techniques of CUSA were described by Honda in 2020 and defined as boring, shoveling, and back-scoring approaches [9]. 

The boring technique consists of the insertion of the tip of the CUSA straight into the parenchyma, aspirating and removing tumoral tissue in a single movement.

Shoveling is the preferred technique by neurosurgeons, using about one-third of the metal tip as a spoon with a single motion.

The back-scoring technique consists of tumor scratching with the end of the metal tip, pursuing the scratching along the dissection line, and pulling out the CUSA handpiece. 

Due to the post-chemotherapy tumor modifications, calcifications can make the shoveling technique dangerous (which is the most common approach for liver and neurosurgical resections) due to the higher discrepancy of solidity between the vascular adventitia and media tunica and calcified parenchyma. The boring technique, consisting of the blind insertion of the CUSA tip straight into the parenchyma to the blood vessel wall, was never used due to the high risk of vascular perforation. 

The back-scoring technique was preferred for the controlled dissection along the cutting line, applying balanced CUSA use with a consistent amount of pressure (Figure 2 and Figure 3). 

Despite the CUSA being designed to stop the parenchyma dissection when the vascular adventitia tunica is encountered, vessel perforation is still possible if the instrument’s tip pressure is maintained on the same point or if a previous vascular dissection was performed. 

To reduce the risk of perforation, a continuous dissection is required to perform the back-scoring technique in safe conditions.

In the case of vascular lesions, the CUSA allows vessel sealing by the stanching principle [9]. Stanching allows bleeding point control using thermal denaturation with compression by the flank of the CUSA electrocautery tip and soft coagulation. The lateral parts of the CUSA cylindrical metallic tip are atraumatic, as longitudinal ultrasonic vibration provides tissue fragmentation and aspiration only at the level of the terminal part of the tip.

This study was approved by the local Institutional Review Board at Bicêtre Hospital and conducted according to the principles of the Declaration of Helsinki. Informed consent was obtained from all patients’ parents. 

## 3. Results

In the analyzed period (2000 to 2018), 43 patients were admitted to Bicêtre Hospital for stage L2 and M neuroblastoma according to the INRGSS classification. 

Among these, 26 children (60.5%) were operated on with the technical support of the CUSA and included in the study. A total of 15 (57.7%) patients were male and the median age at diagnosis was 37 months (range: 0–174 months). 

All the tumors had abdominal localization: 19 (73.1%) were adrenal and 7 (25.9%) were on the median line; 17 (65.4%) had metastasis at the diagnosis, mostly in the bone marrow. All children underwent neoadjuvant chemotherapy according to the International Society of Pediatric Oncology (SIOP) protocols for high-risk neuroblastoma [8], still presenting IDRFs. All patients with adrenal location presented renal pedicle involvement and cava vein encasement. Surgical resections were performed with a median number of four IDRFs [1,2,3,4,5,6] for each patient. Renal pedicle involvement was the most frequent intraoperative finding (n = 23, 88.5%), followed by celiac axis (n = 21, 80.8%), superior mesenteric artery (n = 20, 76.9%), inferior cava vein (n = 12, 46%), aorta (n = 8, 30.8%), iliac vessels (n = 6, 23.1%), and porta hepatis encasement (n = 4, 15.4%).

Seven nephrectomies (26.9%) were required for tumoral infiltration at the level of the renal hilum to obtain a macroscopical radicality in these patients with unfavorable tumor biology. One intestinal resection was performed for tumoral involvement of the duodenum. No massive bleeding occurred during the surgical procedure and 11 patients required transfusion with a median estimated blood loss of 160 mL (80–200).

A total of six patients (23.1%) presented surgical complications (three cases of chylous ascites and three acute renal insufficiencies), but none of them was a grade III Clavien–Dindo requiring re-intervention.

Macroscopic gross resection was performed in 13 patients (50%) and minimal residual (<5 mL, >90% of resection) in the other half of patients.

With a median follow-up of 69.6 months (5.6–140.4), we registered 13 (50%) cases of complete remission and 13 deaths (50%), after local or metastatic recurrence. Among the deceased patients, 11 (84.6%) were metastatic at diagnosis and 3 (23.1%) had N-MYC amplification. 

## 4. Discussion

The surgical treatment of high-risk neuroblastoma remains a controversial and challenging topic in the field of pediatric surgical oncology. The IDRF assessment represents a cornerstone of neuroblastoma management, guiding oncologists and surgeons on the timing and surgical approach of these patients [10,11,12,13,14,15,16]. 

In addition to the higher risk of intra and post-operative surgical morbidity and mortality, the presence of IDRFs are related to a poor oncologic outcome [17,18,19,20,21,22].

Current literature shows that pre-operative chemotherapy has a variable effect on IDRFs, with reported effectiveness of 38.5–81.8% in reducing IDRFs prior to surgery [23,24,25,26]; however, an increase in IDRFs during chemotherapy has also been reported [27].

Recent literature has shown that surgical completeness is related to an increased survival rate in patients with high-risk neuroblastoma, however, complete resection remains challenging due to the presence of IDRFs [28]. The CUSA introduction for IDRF management follows the same principles of hepatobiliary and neurosurgical application, preventing vascular wall lesions during tumor dissection.

Renal pedicle involvement is the most frequent IDRF and nephrectomy for high-risk neuroblastoma is often required to achieve a radical resection with a risk to develop a progressive reduced glomerular function in long-term follow-up [29], but the other abdominal IDRFs related to organ-supply vessels (including celiac axis, porta hepatis, and the superior mesenteric artery) cannot be managed in the same way, as their injury is potentially related to life-threatening events (liver and intestinal failure).

The CUSA technique followed the surgical oncology principle introduced by Kiely [4], replacing the cold knife vascular dissection with an ultrasonic dissection. The cold knife technique introduced the concept of tumor opening straight to the vascular adventitia tunica, respecting the dissection plane, and then using the monopolar electrocautery.

The proposed CUSA technical approach offers several advantages when compared to the monopolar electrocautery and cold knife dissection, allowing a sparing vascular technique working on a dry dissection plane without modification due to the monopolar electricity diffusion to vascular tissues. An additional advantage is a possibility of combining dissection and vascular thermal sealing in case of punctiform perforation. 

During the study period, all patients with neuroblastoma and positive IDRFs were operated on with the CUSA approach, then, in absence of a case-control comparison, we compared our series with the available literature and the largest reported series of patients with similar preoperative characteristics [25,26,28].

Interestingly, in the last decade, NB with IDRFs were defined as ‘unresectable’ [27]. 

The largest surgical series of NB with IDRFs patients published in Europe during our study period showed 8.4% of patients who did not undergo surgery for tumor inoperability, while all patients with IDRF persistency after chemotherapy were operated on in our center.

Resection completeness was differently described in the literature, using the percentage or macroscopic/minimal residual tumor tissue terminology. Comparing our results and the American and European series, with 30% and 24%, respectively, of incomplete resection, the CUSA approach allowed a greater completeness of resection.

The nephrectomy rate was comparable with the current literature, while no normal organ was resected during surgery as reported in the largest American and European studies [28,30].

Data from the patients treated with this technique in our institution showed a low rate of intra-operative and post-operative complications; intra-operative mortality, which is still reported in the largest reported series, never occurred in our series [28,31]. In our cohort, oncology follow-up showed a similar complete remission rate compared to the largest surgical series of patients with high-risk neuroblastoma [28,30].

The application of the CUSA in children has several limitations that are mainly related to the cost-effectiveness of new technology applications in pediatric surgery and the learning curve needed to develop these advances in the pediatric field [31].

Advances in surgical technology are usually applied with a consistent delay when comparing pediatric and adult surgery, which is mainly due to the low case load in pediatrics and related technology costs that are difficult to justify.

Currently, the CUSA is mostly used in pediatric neurosurgery and hepatobiliary surgery [32,33], and not all pediatric hospitals have both of these surgical units.

The multidisciplinary interventional surgical platform and orientation of the team at the national liver surgery and national transplantation center allowed for CUSA access and related technical skills for surgeons. 

CUSA use, like other recent technologies applied in children [34], requires appropriate training. Low-cost simulation models were described to improve CUSA skills [35] and can be considered in the absence of an adequate case load in hepatobiliary surgery. 

The concentration of care is another possible tool to improve the application of new technologies and surgical outcomes in patients with neuroblastoma, as recently reported by van der Steeg and colleagues [36]. The concentration of care has been related to improved surgical outcomes in high-volume hospitals and can be considered to optimize healthcare resources in those centers with experienced pediatric surgical oncology and the availability of modern technologies.

## 5. Conclusions

The proposed technique combines new technology advances and established surgical oncology approaches for neuroblastoma with IDRFs. 

The application of the CUSA could be considered as a complementary safe dissection technique for vascular dissection in the case of high-risk neuroblastoma to provide an extensive resection to improve children’s oncology outcomes. 

## Figures and Tables

**Figure 1 children-10-00089-f001:**
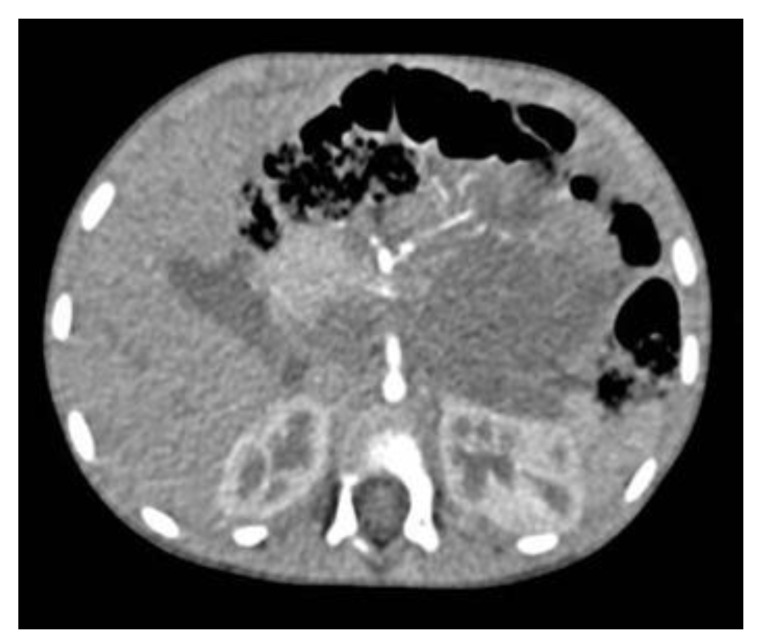
Neuroblastoma computed tomography axial section: encasement of the coeliac trunk and aorta. An encasement is radiologically defined when the vascular surface is > 50% surrounded by the tumor.

**Figure 2 children-10-00089-f002:**
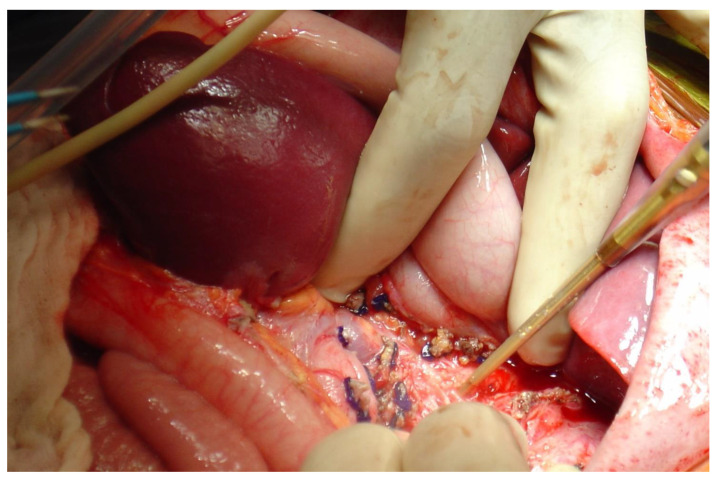
Intraoperative findings: splenic mobilization and cava vein dissection with the CUSA.

**Figure 3 children-10-00089-f003:**
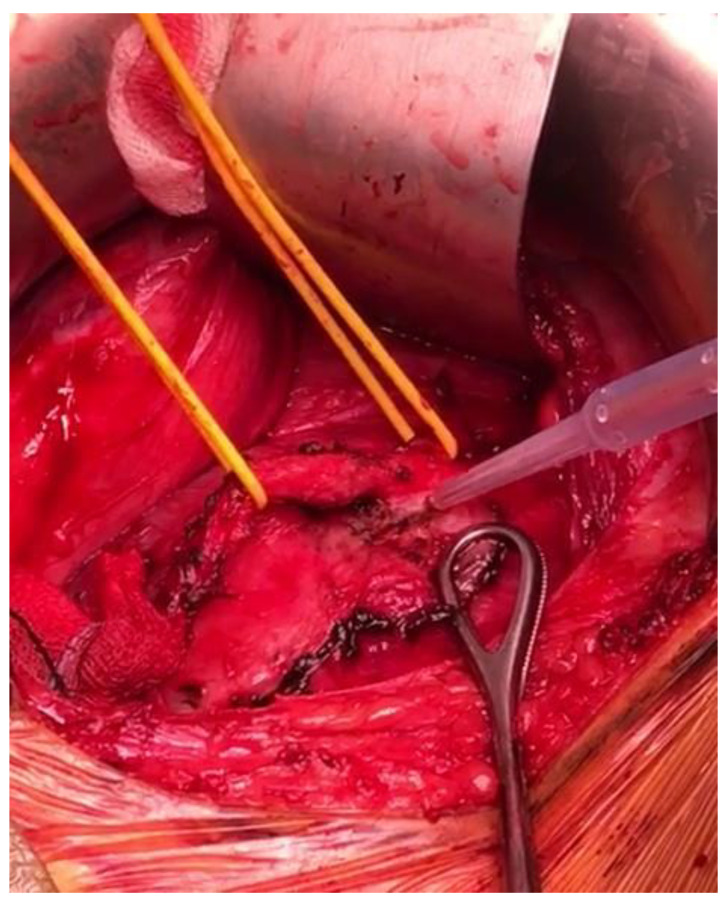
Intraoperative findings: liver mobilization and aorta dissection with the CUSA after neuroblastoma division. Tumor tissues are used as countertraction with Foerster–Ballenger forceps.

## Data Availability

The data are available upon request to the corresponding author.

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
