# Peer review of "The Use of Cavitron Ultrasonic Surgical Aspirator for High-Risk Neuroblastoma with Image-Defined Risk Factors in Children"

_children, 2023, doi:10.3390/children10010089_

Round 1

Reviewer 1 Report

Thank you for a very nicely detailed technique paper.

I understand that the aim of the paper is to describe the technique, but it would be interesting to have a control group operated without CUSA comparing the results on complications and oncological outcomes. This can be a topic for further publications or could be included in this paper.

Author Response

Dear Reviewer 1,

Many thanks for your suggestions. Unfortunately, patients with IDRFs were operated with this technique in the last two decades, the only available control group represented by patients without IDRFs would not represent comparable surgical characteristics. We strongly agree that case/control further studies will be required to consider benefits of this technique, for that reason we compared our results with the largest series reported in literature with similar cohort of patients in discussion section. In addition, we add a proper paragraph in conclusions, discussing the need of further case/control studies to validate our results.

Reviewer 2 Report

The authors have reported their experience on the use of CUSA device in the surgical resection of high-risk neuroblastoma with IDRFs. 26 patients underwent surgical resection with CUSA. They concluded that the use of CUSA is feasible and safe for neuroblastoma resection. 

I believe English is not the authors' native language and recommend that the manuscript should be referred for scientific editing before it should be considered for publication. 

As this paper is focused on the application of the CUSA, its mode of operation, mechanism of parenchyma dissection and its ability to avoid vascular injury should be elaborated with greater detail. Similarly, even though the technqiues of application of CUSA, such as shovelling, boring, "back-scoring" were mentioned, it will be beneficial for the readers who wish to know the various technical steps. Back-scoring that has been preferred by the authors, was not clearly described in the text. The video was inadequate and incomprehensible, as it was not accompanied by elaboration of the technique. It was not apparent from the video presented the effective dissection of the CUSA. 

The following sentences should be restructured to enable better understanding: 

Line 34-35: "infiltrating the adventitious tunic, but displacing and involving it". infiltrating and involving seems to be the same but was used as opposing terms. 

Line 39: "current" should be read "currently"

Line 41: spell out INRGSS in full.

Line 43: "widespread technique". "Widespread" is not a suitable adjective for technique. 

Line 44: "tumor opening", to consider using "splitting the tumor" or "incising the tumor"

Line 46: "technology" should be read as "technological"

Line 48: why is dissection of tumor "controversial"?

Lines 78-84: elaborate "boring", "Shovelling", "back-scoring"

Line 120: "oncology radicality" should be read as "oncological radicality"

Lines 148-150: I query the "concept of nephrectomy for HR neuroblastoma is required to achieve radical resection". This should not be the reason to remove the kidney in HR patients. 

Lines 162-166: Please quote the references on "Data from patients treated with this technique"

Line 169: "difficult availability and surgical exposure needed to master this technique". This should be rephrased. 

Author Response

Dear Reviewer 2,

Thank you for your precious comments and advice. These comments

are all valuable and very helpful for revising and improving our paper.

  1. The revised manuscript was edited by a native English speaker, Isaiah Reeves, from St. Jude School of Biomedical Science.
  2. CUSA mechanism and techniques were more detailed in revised manuscript
  3. The video was removed in revised manuscript for its difficult comprehension, as suggested by reviewer
  4. All the reported sentences were revised according to the reviewer's suggestion, in order to be more clear and comprehensive. In particular for the sentences:

- Line 48: why is dissection of tumor "controversial"?: In revised manuscript it was specified that extended resection is debated in neuroblastoma with favorable biology, while recent international studies reported better oncology outcomes in those tumors with unfavorable biology treated with radical resection

- Elaborate "boring", "Shovelling", "back-scoring": CUSA techniques concept were more detailed in revised manuscript

- I query the "concept of nephrectomy for HR neuroblastoma is required to achieve radical resection". This should not be the reason to remove the kidney in HR patients: The reason to nephrectomy was more detailed in revised manuscript, explaining the relation with radical resection and patients with unfavorable biology as showed by SIOPEN and COG studies (ref 28,28), with an increased survival of those patients underwent radical resection

Round 2

Reviewer 2 Report

The manuscript has been extensively edited and the message is better understood. I would recommend acceptance of the manuscript for publication, but the following minor changes may be required:

1. Line 45: "the most diffused technique" should consider changing it to "the most frequently applied technique"

2. Line 51: "life-threating events" should be changed to "life-threatening events"

3. Line 53: "while complete resection as be shown to be related to better oncology outcomes" should be changed to " while complete resection has been shown to be related to better oncological outcomes"

4. Line 75: "locations, then en bloc" should be changed to "locations. En bloc"

5. Lines 74-76 and lines 77-79 are duplications of and should be rephrased. 

6. Lines 88 and 89: should "whole" be replaced with "hole"?

7. Line 163: "IDRFs" should be "IDRF"

8. Line 167: "IDRFs presence" should be "IDRFs' presence"

9. Line 170: "IDRFs increase" should be "IDRFs' increase"

10. Line 173: "IDRFs presence" should be "IDRFs' presence"

11. Line 176: "IDRFs" should read "IDRF"

12. Line 178: "long terms" should read "long term"

13. Line 180: "their lesion" is better with "their injury"

Author Response

Dear reviewer 2,

Thanks for your suggestions, the thirteen sentences has been corrected in the revised manuscript.